# Flexibility of a large blindly synthetized avatar database for occupational research: Example from the CONSTANCES cohort for stroke and knee pain

Marc Fadel[1]*, Julien Petot[2], Pierre-Antoine Gourraud[3,4], Alexis Descatha[1,5]

1 Univ Angers, CHU Angers, Univ Rennes, Inserm, EHESP, Irset (Institut de recherche en santé, environnement et travail) - UMR_S, IRSET-ESTER, SFR ICAT, CAPTV CDC, Angers, France, 2 Octopize, Nantes, France, 3 Nantes Université, INSERM, CR2TI - Center for Research in Transplantation and Translational Immunology, Nantes, France, 4 Nantes Université, CHU Nantes, Pôle Hospitalo-Universitaire 11: Santé Publique, Clinique des données, INSERM, CIC 1413, Nantes, France, 5 Department of Occupational Medicine, Epidemiology and Prevention, Donald and Barbara Zucker School of Medicine, Hofstra/Northwell, United States of America

* marc.fadel@univ-angers.fr

**Data Availability Statement:** Aggregated data of OR, are given in the S1 Appendix. The dataset used for this study is third-party data and not publicly available to ensure the confidentiality of study

## Abstract

### Objectives

Though the rise of big data in the field of occupational health offers new opportunities especially for cross-cutting research, they raise the issue of privacy and security of data, especially when linking sensitive data from the field of insurance, occupational health or compensation claims. We aimed to validate a large, blinded synthesized database developed from the CONSTANCES cohort by comparing associations between three independently selected outcomes, and various exposures.

### Methods

From the CONSTANCES cohort, a large synthetic dataset was constructed using the avatar method (Octopize) that is agnostic to the data primary or secondary data uses. Three main analyses of interest were chosen to compare associations between the raw and avatar dataset: risk of stroke (any stroke, and subtypes of stroke), risk of knee pain and limitations associated with knee pain. Logistic models were computed, and a qualitative comparison of paired odds ratio (OR) was made.

### Results

Both raw and avatar datasets included 162,434 observations and 19 relevant variables. On the 172 paired raw/avatar OR that were computed, including stratified analyses on sex, more than 77% of the comparisons had a OR difference ≤0.5 and less than 7% had a discrepancy in the statistical significance of the associations, with a Cohen's Kappa coefficient of 0.80.

participants. Nevertheless, they are accessible providing authorizations from the National Data Protection Authority, the INSERM Institutional Review Board and the CONSTANCES' scientific committee. The procedure to access data from the CONSTANCES cohort is available on the web page https://www.constances.fr/conduct-project-ongoing.php. Colleagues interested in replicating our findings are encouraged to contact Marc Fadel. The authors did not have any special access privileges that others would not have.

**Funding:** NO - Include this sentence at the end of your statement: The funders had no role in study design, data collection and analysis, decision to publish, or preparation of the manuscript. Funding. The study is funded by a regional Public Fund (TEC-TOP project): Pays-de-la-Loire Region, Angers Loire Métropole, Univ Angers, CHU Angers. The CONSTANCES Cohort Study was supported and funded by the French National Health Insurance Fund ("Caisse nationale d'assurance maladie", CNAM). The CONSTANCES Cohort Study is an "Infrastructure nationale en Biologie et Santé" and benefits from a grant from the French National Agency for Research (ANR-11-INBS-0002). CONSTANCES is also partly funded by Merck Sharp & Dohme (MSD), AstraZeneca, Lundbeck and L'Oréal through Inserm-Transfert. None of these funding sources had any role in the design of the study, collection and analysis of data or decision to publish.

**Competing interests:** Enter: The authors have declared that no competing interests exist.In addition to the affiliation, Alexis Descatha has received fees from Elsevier Masson as editor-in-chief of the journal "les archives des maladies professionnelles et de l'environnement". Octopize had no role in the design of the study, collection and analysis of data or decision to publish.

## Conclusions

This study shows the flexibility and the multiple usage of a synthetic database created with the avatar method in the particular field of occupational health, which can be shared in open access without risking re-identification and privacy issues and help bring new insights for complex phenomenon like return to work.

## Introduction

The rise of massive data in the past decade has brought heath research to new heights allowing analyses of exhaustive and integrated population data [1]. Though these information technology tools have been developed and have expanded from a long time in other domains [2], their application in health is more recent. Epidemiological studies can include billions of person-year and data thanks to complex linkage of databases [3]. Private and public health insurance databases are particularly interesting because of the sheer number of observations and variables available which allow research on many subjects including description of morbidity and mortality, exploration of etiology and relation between health determinants, study of work and limitations outcomes as wells as medico-economic studies and the impact of policy changes [4,5].

In occupational health research, the development of new modeling approaches like machine learning is bringing new insights in complex phenomenon, like occupational injuries or return-to-work [6–8]. For example, linking data from insurance companies, occupational health units, health care organizations or compensation claims and applying machine learning methods could bring promising results in the field of occupational health research and rehabilitation [6]. However, most of the time, these approaches need a substantial amount of data [9] and some of them (black box models) are not reproducible without the original data which brings out the issue of data sharing [10].

Issues regarding data security and possible re-identification have grown in the public and in governmental agencies. Some studies managed to re-identify databases with great accuracy which highlights the unsatisfactory reality for data anonymization [11,12]. Data related to the occupational medical files are of particular concern in terms of security and confidentiality as they deal with both health data (medical confidentiality) and data related to the workplace (professional confidentiality). Furthermore, the latest guidelines in France recommend collecting information on sensitive topics like medical fitness for work, occupational exposures, addictions, psychiatric diseases, vaccinations etc [13,14]. With that in mind, the General Data Protection Regulation (GDPR) imposed harsh conditions for defining anonymous data and proposed several recommendations to evaluate the robustness of an anonymization process [15].

Synthetic data generation methods are powerful tools that can generate a dataset different from the original while retaining its statistical properties [16,17]. Several methods exist but they often require knowing the underlying hypotheses on the variables' relation in specific analyses [18], and rarely allows a true quantification of the re-identification risk [17]. The avatar method has been suggested for generating synthetic datasets and has proven to be effective in specific analyses in terms of statistical properties as well as ensuring data privacy [19]. However, this method has yet to be tested in the context of a large cohort dataset with few hypotheses for possible statistical analyses when generating the synthetic dataset.

Therefore, the aim of this study was to create a large synthetic database and evaluate associations between three selected outcomes and several relevant variables using data from the CONSTANCES cohort.

## Methods

The initial database used in this study originated from data of the CONSTANCES cohort, a French population-based cohort created in 2012. The design and characteristics of the cohort are detailed in another paper [20]. In short, approximately 200,000 randomly selected adults aged between 18 to 69 years and affiliated with the National Social Security system (more than 80% of the French population) make up the cohort. Each participant completed several self-administered baseline questionnaires and underwent a health screening by medical professionals at Social Security affiliated centers and completed self-administered surveys. These questionnaires and examinations allowed having data on demographic, lifestyle habits, occupational exposures, comorbidities including joint pain and daily limitations. The first case of stroke during the period of data collection (2012–2018) was also available thanks to a linkage between the cohort and the French National Health Insurance records.

The initial database consisted of 162,434 observations and 280 variables that had been previously obtained for other studies [21,22]. The Avatar method, was used to create a synthetic dataset ("avatar dataset") based on the original data ("raw dataset"). Details on how the Avatar methods works are available in another paper [19]. The team in charge of synthetizing the raw dataset was blinded to the analysis which would be run using variables from the raw dataset. The only information was a list of 22 variables of interest (S1 Appendix) to which a weight of 10 was added. The parameter chosen for the creation of the avatar dataset were k = 20 (number of neighbors) and ncp = 20 (number of components for the search of neighbors). Neighbors correspond to the closest individuals from a data point using the Euclidian distance in the multidimensional projection space.

Three main analyses were carried out in both the raw dataset and the avatar dataset. Bivariate logistic or multinomial models were computed to evaluate the associations considered. The first main analysis assessed the association between stroke and relevant risk factors including age (continuous), sex (male, female), body mass index (continuous), smoking (number of pack years, i.e. the number of packs smoked per day multiplied by the number of years the person has smoked), history of diagnosed dyslipidemia (no, yes), diabetes (no, yes), high blood pressure (no, yes), occupational status (high skilled white-collar jobs, Self-owner/Chief Executive Officer/Professional jobs, low skilled white-collar jobs, blue-collar workers) and long working hours exposure duration ($<10$ years, $\geq 10$ years). Long working hour was defined in the CONSTANCES cohort as working time greater than 10 hours a day for more than 50 days per year. The outcome was an occurrence of stroke (no, yes all subtypes) or subtype of strokes (no, ischemic stroke, hemorrhagic stroke). The second main analysis assessed the association between knee pain and relevant risk factors including age, sex, body mass index, occupational status, sedentary lifestyle (no, yes) and high rating on Borg Rating of Perceived Exertion Scale ($<13$, $\geq 13$). The outcome was considered in three categories: no/low pain, moderate pain and severe pain. The third main analysis assessed the effect of knee pain (as defined in the 2nd main analysis) on several reported limitations: daily life limitations because of health problems (no, yes), daily life limitations because of pain (no, yes), limitations for climbing stairs (no, yes), limitations for walking (no, yes) and limitations for carrying 5kg (no, yes). For all the main analyses, stratified analyses for the male and female subgroups were carried out. Variables selected for the three main analyses included both native variables, i.e. variables that were not modified (age, body mass index, diabetes diagnosed) and "reconstructed" variables, i.e. either by combining several categories together (occupational status, stroke subtypes, long working hours) or by combining several variables together (smoking, sedentary lifestyle, knee pain).

A qualitative approach was used to compare the raw/avatar paired odds-ratios (ORs) and their 95% confidence interval as the aim was to obtain clinically comparable associations and

not necessarily associations that were statistically not different. A Cohen's kappa coefficient was calculated to measure the reliability of significance of the ORs based on the raw dataset or avatar dataset. Analyses were performed using R (version 4.2, packages "CompareGroups, EpiDisplay").

All study participants provided written informed consent prior to enrollment. The study had institutional review board approval for research on human subjects (Commission nationale de l'informatique et des libertés n˚ 910486, "Comité consultatif sur le traitement de l'information en matière de recherche n˚10.628).

## Results

Both raw dataset and avatar dataset had 162,434 observations and the same number of variables. A descriptive table of the two datasets can be found in Table 1.

Differences between the Avatar and Raw data sets in terms of the percentage distribution of responses for the various risk factors were mostly small ($\leq$2%), with only one category, the "Current/former smoker <30" which had a moderate difference (>5% but $\leq$10%). On the 172 raw/avatar paired ORs calculated (Table 1 and S1 Appendix), 77% had a difference $\leq$0.5. OR were closer to each other for smaller effect magnitude (Fig 1).

There was a strong agreement when considering the reliability of a significant association, with a Cohen's kappa coefficient of 0.80. Among the 13 raw/avatar paired ORs with a discrepancy in the significance of the associations, only one had an OR difference >0.5 and another had an opposite significant association.

## Discussion

This study shows the flexibility and the multiple usage of a synthetic database created with the avatar method. Protection of data is a serious issue which rightfully impedes data sharing. The aim of this study was not to prove the statistical superiority of the avatar methods, but to test it in a blinded exploratory context of a large cohort dataset with minimum assumptions on the analyses that would be carried out. Though collaboration and open access data have become key aspects of modern research, stealing data for economic gains has become more frequent, and health facilities are not immune to this threat [23]. The avatar method seems to be a promising tool for protecting sensitive data and for both specific analyses and exploration analysis. The avatar version of an exhaustive database could be used by several researchers to study multiple outcomes and factors and share their hypotheses before formulating more precise statistical and etiological models [18]. Large, anonymized databases linking sensitive data are relevant in the case of data pertaining to work, health, or compensation data, and in the case of data sharing and open access, especially for black-box machine learning models that cannot be shared with the training datasets. In the rehabilitation field, powerful machine learning models trained on large datasets could be promising tools for on field practitioners, and not just research, if they become available without compromising the original participants' data privacy.

Possible application examples could include presentation of data to employers and workers on sensitive issues (like addictions), sharing databases used for analyses (transparency), sharing databases for multicentric analyses (e.g. across occupational health occupational health units) with possibilities of international projects, or sharing databases that were used to train machine learning models.

The avatar associations seemed to overestimate raw associations $\geq$2 (Fig 1). One explanation could be linked to the behavior of the avatar method. By design, the Avatar method tends to center individuals to local barycenters. With a parameter k = 20, the avatar method could

**Table 1. Distribution of variables in the raw and avatar dataset, crude odds ratios for the occurrence of stroke (Stroke analysis), and crude odds ratios for reported limitations, across selected risk factors.**

| | Avatar | Raw | Crude Odds Ratio Avatar dataset | Crude Odds Ratio Raw dataset |
|---|---|---|---|---|
| Age (Years) | 47.8 (12.3) | 47.8 (13.1) | 1.07 (1.07–1.08) | 1.07 (1.06–1.07) |
| Sex | | | | |
| Female | 85865 (52.9%) | 86132 (53.1%) | Ref | Ref |
| Male | 76422 (47.1%) | 76154 (46.9%) | 2.15 (1.85–2.51) | 1.93 (1.66–2.25) |
| Body mass index (kg/m$^2$) | 24.9 (3.31) | 25.0 (4.48) | 1.10 (1.08–1.12) | 1.05 (1.04–1.07) |
| Dyslipidemia diagnosed | | | | |
| No | 149545 (92.1%) | 148550 (91.5%) | Ref | Ref |
| Yes | 12889 (7.93%) | 13884 (8.55%) | 5.63 (4.81–6.57) | 5.41 (4.63–6.30) |
| Smoking (Pack.Years) | | | | |
| Not smoker | 72083 (48.6%) | 89251 (54.9%) | Ref | Ref |
| Current/former smoker <30 pack-years | 74090 (49.9%) | 66789 (41.1%) | 1.19 (1.02–1.41) | 1.07 (0.91–1.25) |
| Current/former smoker ≥30 pack-years | 2289 (1.54%) | 6394 (3.94%) | 7.46 (5.58–9.82) | 3.67 (2.90–4.60) |
| Diabetes diagnosed | | | | |
| No | 161023 (99.1%) | 160270 (98.7%) | Ref | Ref |
| Yes | 1411 (0.87%) | 2164 (1.33%) | 4.82 (3.24–6.89) | 3.41 (2.34–4.79) |
| High blood pressure diagnosed | | | | |
| No | 144509 (89.0%) | 143516 (88.4%) | Ref | Ref |
| Yes | 17925 (11.0%) | 18918 (11.6%) | 4.40 (3.78–5.13) | 4.52 (3.88–5.24) |
| Occupation | | | | |
| High-skilled white-collar jobs | 45481 (30.5%) | 43830 (30.0%) | Ref | Ref |
| Self-employed/Chief Executive Officer, Professional jobs | 49436 (33.2%) | 49623 (34.0%) | 1.15 (0.95–1.39) | 1.16 (0.95–1.42) |
| Low-skilled white-collar jobs | 39452 (26.5%) | 37742 (25.8%) | 0.85 (0.68–1.06) | 0.98 (0.78–1.22) |
| Blue-collar jobs | 14534 (9.76%) | 14955 (10.2%) | 1.41 (1.08–1.81) | 1.73 (1.35–2.20) |
| Long working hours | | | | |
| <10 years | 142198 (87.5%) | 141361 (87.0%) | Ref | Ref |
| ≥10 years | 20236 (12.5%) | 21073 (13.0%) | 1.53 (1.26–1.84) | 1.72 (1.43–2.05) |
| Stroke | | | | |
| No | 161704 (99.6%) | 161697 (99.5%) | / | / |
| Yes | 730 (0.45%) | 737 (0.45%) | / | / |
| Stroke subtypes | | | | |
| No Stroke | 161751 (99.6%) | 161697 (99.5%) | / | / |
| Ischemic stroke | 503 (0.31%) | 534 (0.33%) | / | / |
| Hemorrhagic stroke | 180 (0.11%) | 203 (0.12%) | / | / |
| Sedentary lifestyle | | | | |
| No | 71625 (44.1%) | 71277 (43.9%) | / | / |
| Yes | 90809 (55.9%) | 91157 (56.1%) | / | / |
| Borg scale | | | / | / |
| <13 | 133742 (82.3%) | 130114 (80.1%) | / | / |
| ≥13 | 28692 (17.7%) | 32320 (19.9%) | / | / |
| Daily life limitations because of health problems | | | | |
| No | 113538 (69.9%) | 109655 (67.5%) | Ref | Ref |
| Yes | 48896 (30.1%) | 52779 (32.5%) | *1.82 (1.78–1.86) | *1.73 (1.69–1.77) |
| | | | †6.79 (6.57–7.02) | †5.48 (5.32–5.65) |
| Daily life limitations because of pain | | | | |
| No | 147493 (90.8%) | 145594 (89.6%) | Ref | Ref |
| Yes | 14941 (9.20%) | 16840 (10.4%) | *1.89 (1.81–1.97) | *1.80 (1.73–1.88) |

(*Continued*)

**Table 1.** (Continued)

| | Avatar | Raw | Crude Odds Ratio Avatar dataset | Crude Odds Ratio Raw dataset |
|---|---|---|---|---|
| | | | †7.86 (7.52–8.23) | †6.78 (6.51–7.07) |
| Limitations for climbing stairs | | | | |
| No | 154280 (95.0%) | 152306 (93.8%) | Ref | Ref |
| Yes | 8154 (5.02%) | 10128 (6.24%) | *1.91 (1.79–2.04) | *1.83 (1.72–1.95) |
| | | | †12.5 (11.8–13.3) | †11.8 (11.2–12.5) |
| Limitations for walking | | | | |
| No | 155736 (95.9%) | 153704 (94.6%) | Ref | Ref |
| Yes | 6698 (4.12%) | 8730 (5.37%) | *2.10 (1.96–2.26) | *1.72 (1.61–1.84) |
| | | | †13.1 (12.3–14.1) | †10.6 (10.0–11.2) |
| Limitations for carrying 5kg | | | | |
| No | 152492 (93.9%) | 149489 (92.0%) | Ref | Ref |
| Yes | 9942 (6.12%) | 12945 (7.97%) | *1.79 (1.70–1.89) | *1.46 (1.39–1.53) |
| | | | †8.50 (8.06–8.97) | †5.47 (5.23–5.72) |
| Knee pain | | | | |
| No/low pain | 77836 (47.9%) | 79328 (48.8%) | / | / |
| Moderate pain | 63725 (39.2%) | 56329 (34.7%) | / | / |
| Severe pain | 20873 (12.9%) | 26777 (16.5%) | / | / |

* OR for those with moderate knee pain

†OR for those with severe knee pain.

create some local clusters. This could magnify the computed odds ratio. A similar behavior was found for the effect size in the case of low k value [19]. In the case of exploratory analyses, it may not be a liability as the aim is to identify clinically relevant associations, meaning that an effect size greater than 2 would be considered as a "strong" association.

The main limitation of this study is that only one database was used to compare the associations of the avatar and raw databases. Therefore, generalizing the behavior of avatar associations is difficult, especially since it could also depend on the parameter selected for the creation of the avatar dataset. However, the multiplicity of the analyses and the variables used as well as the size of the raw database gives us confidence in the reproducibility of our results. Another limit of the avatar method is that the creation of the avatar dataset requires the solicitation of a third-party software, is not available in open access. Lastly, only a qualitative comparison of the associations since the aim of this study was to obtain associations that were clinically close. Most of the studies compare the overall of confidence intervals [24], though this method has limitations in the case of large dataset with small confidence intervals, as was the case our analysis of knee pain, which had close OR with non-overlapping confidence intervals (Table 1). The partially blinded creation of the avatar dataset is the main strength of this study. Though the data analysts knew about a list of variables of interest (see S1 Appendix), they did not know which analyses would be conducted and had to synthesize a very large database (280 variables). Another strong point is the dataset which originates from a large population-based cohort allowing exploring multiple aspects of health at the same time, from rare events like stroke to common health issues like knee pain and daily life limitations.

To conclude, the avatar method for creating synthesized data seems to be a promising tool for flexible, exploratory analyses of large databases, especially for data sharing and open access, though further research in other large databases will be needed to confirm these findings.

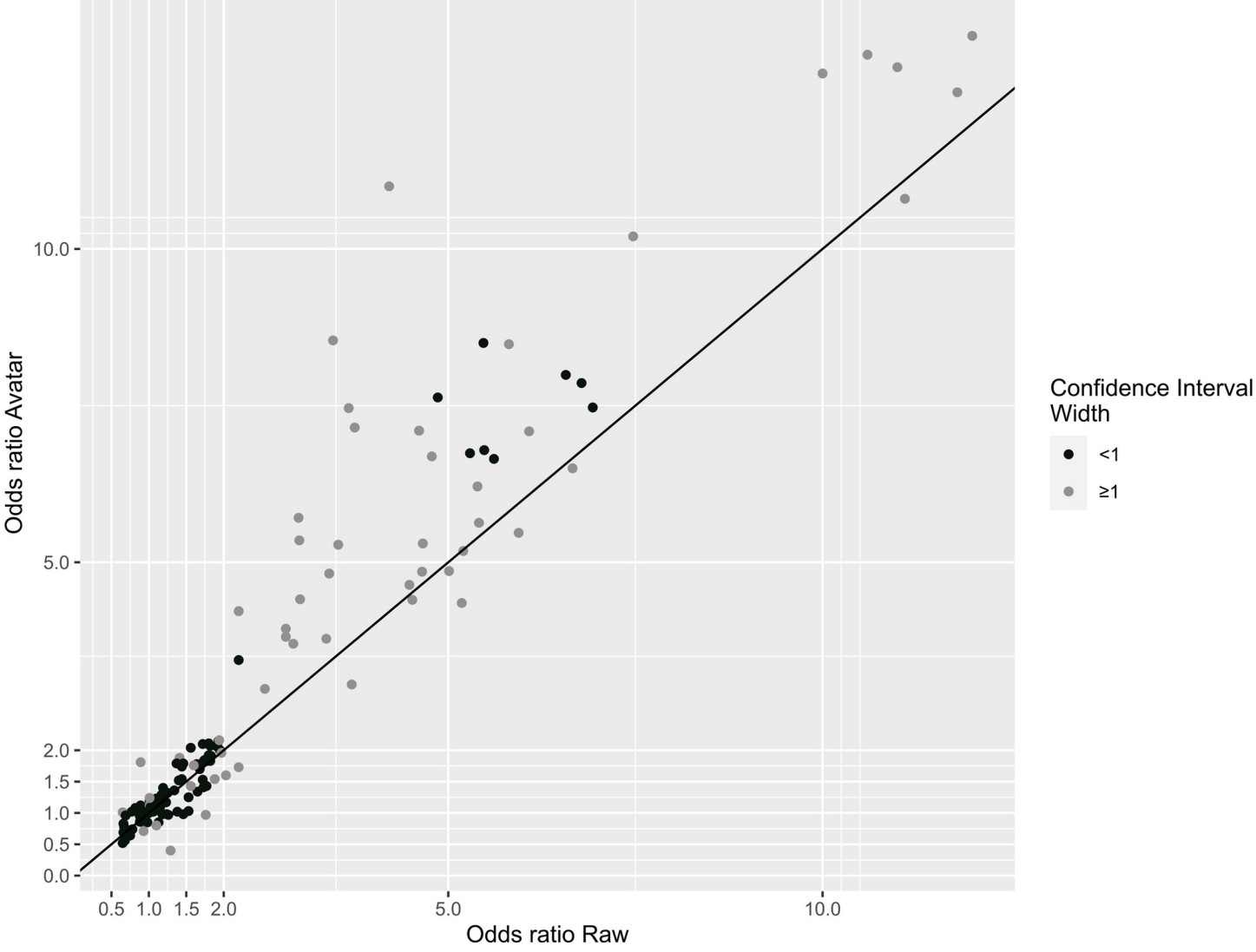

**Fig 1. Scatter plot of paired raw/avatar odds ratio by their confidence interval mean width.**

## Supporting information

**S1 Appendix. List of variables to which weights were assigned and Table of all raw/avatar pair odds ratios.**
(PDF)

## Author Contributions

**Conceptualization:** Marc Fadel, Pierre-Antoine Gourraud, Alexis Descatha.

**Data curation:** Marc Fadel, Julien Petot, Alexis Descatha.

**Formal analysis:** Marc Fadel, Julien Petot, Alexis Descatha.

**Funding acquisition:** Alexis Descatha.

**Methodology:** Marc Fadel, Pierre-Antoine Gourraud.

**Supervision:** Alexis Descatha.

**Writing – original draft:** Marc Fadel.

**Writing – review & editing:** Julien Petot, Pierre-Antoine Gourraud, Alexis Descatha.

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
