## [Editor Report · Decision Letter 0]

30 Oct 2023

PONE-D-23-2774Flexibility of a large blindly synthetized avatar database for occupational research: example from the CONSTANCES cohort for stroke and knee painPLOS ONE

Dear Dr. Fadel,

Thank you for submitting your manuscript to PLOS ONE. After careful consideration, we feel that it has merit but does not fully meet PLOS ONE’s publication criteria as it currently stands. Therefore, we invite you to submit a revised version of the manuscript that addresses the points raised during the review process.

We look forward to receiving your revised manuscript.

Kind regards,

Mir Ali, Ph.D

Guest Editor

PLOS ONE

Journal Requirements:

 "NO - Include this sentence at the end of your statement: The funders had no role in study design, data collection and analysis, decision to publish, or preparation of the manuscript.

Funding. The study is funded by a regional Public Fund (TEC-TOP project): Pays-de-la-Loire Region, Angers Loire Métropole, Univ Angers, CHU Angers. The CONSTANCES Cohort Study was supported and funded by the French National Health Insurance Fund (“Caisse nationale d’assurance maladie”, CNAM). The CONSTANCES Cohort Study is an “Infrastructure nationale en Biologie et Santé” and benefits from a grant from the French National Agency for Research (ANR-11-INBS-0002). CONSTANCES is also partly funded by Merck Sharp & Dohme (MSD), AstraZeneca, Lundbeck and L’Oréal through Inserm-Transfert. None of these funding sources had any role in the design of the study, collection and analysis of data or decision to publish."

6. Please upload a copy of Supporting Information Figure/Table/etc. Supplemental materials which you refer to in your text on page 4, 7 and 9.

**Additional Editor Comments:**

This is original research carried out to assess the utility of the 'Avatar' anonymization technique to create a synthetic data set that retains the statistical characteristics of the raw data set while preventing re-identification of individuals in the raw data set. The study is conducted well and the manuscript is an easy read. References 17 and 18 are important to get a better understanding of the Avatar methodology and the CONSTANCES database.

There are some minor edits that I will recommend. These include the following:

Page 3, paragraph 2, line 5: occupational ‘health research’ and rehabilitation

Page 3, paragraph 3, line 1: that ‘applies’ to occupational health

Page 3, paragraph 3, line 3: managed to “re-identify”

Page 3, paragraph 3, line 8: Possibly meant “properties” and not “proprieties”

Page 6 Table 1 title: Isn’t knee pain an outcome? It is labeled here as a risk factor

Page 7 second to last line: There is a reference for 'Supplemental Material', but no material has been provided that satisfies this reference

However, the major/important shortcoming in this version is that the supplemental material is missing. The authors mention "The dataset analyzed is available in the Supplemental materials" but I could not find the dataset. What was available included two .pdf documents:

2021-186 anglais

2021-186

Once the supplemental materials are received, we can proceed with the next step.

---

## [Author Response · Author response to Decision Letter 0]

30 Nov 2023

Journal Requirements:

R1.1. Files were reviewed and corrected to fit the PLOS ONE’s style requirements. File were renamed accordingly.

2. We note that the grant information you provided in the ‘Funding Information’ and ‘Financial Disclosure’ sections do not match.When you resubmit, please ensure that you provide the correct grant numbers for the awards you received for your study in the ‘Funding Information’ section.

R1.2. Grant information was updated in the submission system was updated to match the one in the manuscript. The funding information was removed from the manuscript accordingly to PLOS ONE’s requirements. Not all funding had specific grant numbers (notably for the funding of the CONSTANCES cohort).

 "NO - Include this sentence at the end of your statement: The funders had no role in study design, data collection and analysis, decision to publish, or preparation of the manuscript.

Funding. The study is funded by a regional Public Fund (TEC-TOP project): Pays-de-la-Loire Region, Angers Loire Métropole, Univ Angers, CHU Angers. The CONSTANCES Cohort Study was supported and funded by the French National Health Insurance Fund (“Caisse nationale d’assurance maladie”, CNAM). The CONSTANCES Cohort Study is an “Infrastructure nationale en Biologie et Santé” and benefits from a grant from the French National Agency for Research (ANR-11-INBS-0002). CONSTANCES is also partly funded by Merck Sharp & Dohme (MSD), AstraZeneca, Lundbeck and L’Oréal through Inserm-Transfert. None of these funding sources had any role in the design of the study, collection and analysis of data or decision to publish."

R1.3. We have updated the founding statement accordingly.

R1.4. Aggregated data of OR, are given in the S1 Appendix. Individual data are not available as open data due to the restriction by our national regulatory agency (“Commission nationale de l’informatique et des libertés”, n°910486). However, the CONSTANCES cohort is “an open epidemiological laboratory” and access to study protocols and data is available on justified request. We have updated the Data Availability statement.

R1.5. The ethical statement was moved to the Methods section as instructed.

6. Please upload a copy of Supporting Information Figure/Table/etc. Supplemental materials which you refer to in your text on page 4, 7 and 9.

R1.6. We apologize for the oversight. Supplemental materials have been added. They contain a table with all the OR calculated for the study and the variables that had were given weight in the creation of the avatar dataset.

Additional Editor Comments:

This is original research carried out to assess the utility of the 'Avatar' anonymization technique to create a synthetic data set that retains the statistical characteristics of the raw data set while preventing re-identification of individuals in the raw data set. The study is conducted well and the manuscript is an easy read. References 17 and 18 are important to get a better understanding of the Avatar methodology and the CONSTANCES database.

There are some minor edits that I will recommend. These include the following:

Page 3, paragraph 2, line 5: occupational ‘health research’ and rehabilitation

R2.1. We have corrected this mistake.

Page 3, paragraph 3, line 1: that ‘applies’ to occupational health

R2.2. We have corrected this mistake.

Page 3, paragraph 3, line 3: managed to “re-identify”

R2.3. We have corrected this mistake.

Page 3, paragraph 3, line 8: Possibly meant “properties” and not “proprieties”

R2.4. We have corrected this mistake.

Page 6 Table 1 title: Isn’t knee pain an outcome? It is labeled here as a risk factor

R2.5. Actually, knee pain in this analysis is a risk factor since the models evaluated the association between having knee pain and the risk of daily life limitations (outcomes). For example, having moderate knee pain was associated with higher risk of limitations for carrying 5kg (OR = 1.79 in the avatar dataset and OR = 1.46 in the raw dataset).

Page 7 second to last line: There is a reference for 'Supplemental Material', but no material has been provided that satisfies this reference. However, the major/important shortcoming in this version is that the supplemental material is missing. The authors mention "The dataset analyzed is available in the Supplemental materials" but I could not find the dataset. What was available included two .pdf documents:

2021-186 anglais

2021-186

Once the supplemental materials are received, we can proceed with the next step.

R2.6. We apologize for the oversight. Supplemental materials have been added. They contain a table with all the OR calculated for the study and the variables that had were given weight in the creation of the avatar dataset.

R2.7. It seems I can’t connect to PACE (the login button doesn’t work for me?). We have changed the Figure to Tiff.

---

## [Decision Letter · Decision Letter 1]

11 Mar 2024

PONE-D-23-27743R1Flexibility of a large blindly synthetized avatar database for occupational research: example from the CONSTANCES cohort for stroke and knee painPLOS ONE

Dear Dr. Fadel,

Thank you for submitting your manuscript to PLOS ONE. After careful consideration, we feel that it has merit but does not fully meet PLOS ONE’s publication criteria as it currently stands. Therefore, we invite you to submit a revised version of the manuscript that addresses the points raised during the review process.

We look forward to receiving your revised manuscript.

Kind regards,

Mir Ali, Ph.D

Guest Editor

PLOS ONE

Journal Requirements:

Reviewers' comments:

Reviewer's Responses to Questions

**Comments to the Author**

1. If the authors have adequately addressed your comments raised in a previous round of review and you feel that this manuscript is now acceptable for publication, you may indicate that here to bypass the “Comments to the Author” section, enter your conflict of interest statement in the “Confidential to Editor” section, and submit your "Accept" recommendation.

Reviewer #1: (No Response)

Reviewer #2: All comments have been addressed

2. Is the manuscript technically sound, and do the data support the conclusions?

Reviewer #1: Partly

Reviewer #2: Yes

3. Has the statistical analysis been performed appropriately and rigorously? 

Reviewer #1: No

Reviewer #2: Yes

4. Have the authors made all data underlying the findings in their manuscript fully available?

Reviewer #1: Yes

Reviewer #2: Yes

5. Is the manuscript presented in an intelligible fashion and written in standard English?

Reviewer #1: No

Reviewer #2: Yes

6. Review Comments to the Author

Reviewer #1: Thank you for the opportunity to review your manuscript titled "Flexibility of a large blindly synthesized avatar database for occupational research: example from the CONSTANCES cohort for stroke and knee pain."

Overall, the manuscript presents an intriguing study aiming to validate a large, blinded synthesized database. However, there are several points that require clarification and further elaboration:

The manuscript would benefit from a more detailed explanation of the statistical methods employed and a thorough interpretation of the results. It remains unclear whether the proposed method is superior to existing approaches, making it challenging to assess the validity of the conclusions drawn.

The data presented in Figure 1 indicate that the avatar odds ratio is above 2, suggesting a magnified effect. However, this aspect lacks explanation within the manuscript. Please provide further insights or discuss possible reasons for this observation.

While the study mentions addressing privacy concerns in occupational health research, concrete examples of practical application are lacking. It would be beneficial to include specific instances or scenarios where the proposed approach can effectively address privacy issues in practice.

Reviewer #2: This paper is very important in the era of open access to medical data obtained in various centers. Conclusions drawn from such large datasets may be crucial for the development of e.g. new therapies, rehabilitation, etc, what Authors discussed in the Introduction. At the same time, the open access formula creates a risk of re-identification of individual data. Therefore, the use of the Avatar method has a great future, especially in the light of the results presented in this manuscript, which are worth publishing.

I just have a few minor questions and comments.

What does neighbors mean in the text, it is not clear for me

Please describe what the authors define as "long working hours" - how many hours a day, a week, a month?

In the table 1 it is row: Current/former smoker <30 – no explanation whether 30 means e.g. 30 cigarette a day?

I don’t understand why wasn't the risk calculation for, Sedentary lifestyle and Borg scale.

Although some authors deny the connection between sedentary lifestyle and stroke, many papers confirm it. Occupational physical effort may also be a risk factor for stroke. This current analysis could provide an additional argument for or against these relationships.

Please take a look on, for example, these papers

• Bahls M, Leitzmann MF, Karch A, Teumer A, Dörr M, Felix SB, Meisinger C, Baumeister SE, Baurecht H. Physical activity, sedentary behavior and risk of coronary artery disease, myocardial infarction and ischemic stroke: a two-sample Mendelian randomization study. Clin Res Cardiol. 2021 Oct;110(10):1564-1573. doi: 10.1007/s00392-021-01846-7

• Young DR, Hivert MF, Alhassan S, Camhi SM, Ferguson JF, Katzmarzyk PT, Lewis CE, Owen N, Perry CK, Siddique J, Yong CM; Physical Activity Committee of the Council on Lifestyle and Cardiometabolic Health; Council on Clinical Cardiology; Council on Epidemiology and Prevention; Council on Functional Genomics and Translational Biology; and Stroke Council. Sedentary Behavior and Cardiovascular Morbidity and Mortality: A Science Advisory From the American Heart Association. Circulation. 2016 Sep 27;134(13):e262-79. doi: 10.1161/CIR.0000000000000440.

• García-Cabo C, López-Cancio E. Exercise and Stroke. Adv Exp Med Biol. 2020;1228:195-203. doi: 10.1007/978-981-15-1792-1_13.

• Hall C, Heck JE, Sandler DP, Ritz B, Chen H, Krause N. Occupational and leisure-time physical activity differentially predict 6-year incidence of stroke and transient ischemic attack in women. Scand J Work Environ Health. 2019 May 1;45(3):267-279. doi: 10.5271/sjweh.3787.

If Authors don’t have a possibilities to perform additional analysis, they should explain the reasons

In Discussion I found double use of „are more” (stealing data for economic gains are more are more frequent, and health facilities are no longer spared)

7. PLOS authors have the option to publish the peer review history of their article (what does this mean?). If published, this will include your full peer review and any attached files.

Reviewer #1: No

Reviewer #2: No

---

## [Author Response · Author response to Decision Letter 1]

22 Apr 2024

Journal Requirements:

0.1. Please review your reference list to ensure that it is complete and correct. If you have cited papers that have been retracted, please include the rationale for doing so in the manuscript text, or remove these references and replace them with relevant current references. Any changes to the reference list should be mentioned in the rebuttal letter that accompanies your revised manuscript. If you need to cite a retracted article, indicate the article’s retracted status in the References list and also include a citation and full reference for the retraction notice.

R0.1. The reference list has been checked.

Reviewer #1. Thank you for the opportunity to review your manuscript titled "Flexibility of a large blindly synthesized avatar database for occupational research: example from the CONSTANCES cohort for stroke and knee pain."

Overall, the manuscript presents an intriguing study aiming to validate a large, blinded synthesized database. However, there are several points that require clarification and further elaboration:

1.1. The manuscript would benefit from a more detailed explanation of the statistical methods employed and a thorough interpretation of the results. It remains unclear whether the proposed method is superior to existing approaches, making it challenging to assess the validity of the conclusions drawn. 

R1.1. We understand the point raised by the reviewer, however, the aim of the paper was not to prove the superiority of the avatar method, but to test it in a specific context. Indeed, synthetic methods requires making assumptions on the distribution and the relation between variables of interest. In this work, we tried to create a synthetic dataset from a large database in the case where the team creating the synthetic dataset was blinded to the analyses that were done by the team in charge of the statistical analyses. Thus, this study represents a practical example of an exploratory analysis based on a cohort dataset with many variables and observations. Such example could be a first step and allow researchers and practitioners to make a first “crude” analysis on a dataset while minimizing the risk of reidentification in case of data leaking. A confirmatory analysis would of course have to be made on the real dataset, but would take less time thanks to the pre-analyses of the avatar dataset. 

We have clarified the aim of the study in the discussion part of the manuscript:

p8 “The aim of the study was not to prove the statistical superiority of the avatar methods, but to test it in a blinded exploratory context of a large cohort dataset with minimum assumptions on the analyses that would be carried out”

1.2. The data presented in Figure 1 indicate that the avatar odds ratio is above 2, suggesting a magnified effect. However, this aspect lacks explanation within the manuscript. Please provide further insights or discuss possible reasons for this observation.

R1.2. At this stage of exploration of the avatar dataset, it is difficult to explain all of the

observed behaviors. One explanation could be linked to the behavior of the avatar method. By design, the Avatar method tends to center individuals to local barycenters. 

With a parameter k=20, the avatar method could create some local clusters. This could

magnify the computed odds ratio. 

We have added precision in the manuscript on this point p9.

1.3. While the study mentions addressing privacy concerns in occupational health research, concrete examples of practical application are lacking. It would be beneficial to include specific instances or scenarios where the proposed approach can effectively address privacy issues in practice.

R1.3. We have followed the reviewer’s advice and added practical example in the manuscript as well as details on the peculiarity of the occupational medical file.

In the introduction p3 “Likely, data related to the occupational medical files are of concern in terms of security and confidentiality as they deal with both health data (medical confidentiality) and data related to the workplace (professional confidentiality). Thus, the latest guidelines in France recommend collecting information on sensitive topics like medical fitness for work, occupational exposure but also addictions, psychiatric diseases, vaccinations etc”.

In the discussion p9 “Possible application examples could include restitution of statistical analyses to employers and workers on sensitive issues (like addictions), sharing databases used for analyses to employers and workers (transparency), sharing databases for multicentric analyses (like between occupational health units) with possibilities of international projects, Or sharing databases that were used to train machine learning models”.

Reviewer #2: This paper is very important in the era of open access to medical data obtained in various centers. Conclusions drawn from such large datasets may be crucial for the development of e.g. new therapies, rehabilitation, etc, what Authors discussed in the Introduction. At the same time, the open access formula creates a risk of re-identification of individual data. Therefore, the use of the Avatar method has a great future, especially in the light of the results presented in this manuscript, which are worth publishing.

I just have a few minor questions and comments.

2.1. What does neighbors mean in the text, it is not clear for me

R2.1. In the avatar methods, individual observations are projected into a complete multidimensional space. So, all individuals have numerical coordinates, and we can compute distances between individuals. Thus, by neighbors, we mean the closest individuals from a data point using the Euclidian distance in the projection space.

We have clarified this point in the methods p5: “Neighbors correspond to the closest individuals from a data point using the Euclidian distance in the multidimensional projection space”.

2.2. Please describe what the authors define as "long working hours" - how many hours a day, a week, a month?

R2.2. Long working hours was defined in the CONSTANCES cohort as a working time greater than 10 hours a day for more than 50 days per year. The duration of exposure to long working hours was also collected. 

We added this definition in the methods:

p5 “Long working hour was defined in the CONSTANCES cohort as working time greater than 10 hours a day for more than 50 days per year.”

2.3 In the table 1 it is row: Current/former smoker <30 – no explanation whether 30 means e.g. 30 cigarette a day?

R2.3. Smoking was quantified in pack-years. The information was added in the table.

2.4. I don’t understand why wasn't the risk calculation for, Sedentary lifestyle and Borg scale. Although some authors deny the connection between sedentary lifestyle and stroke, many papers confirm it. Occupational physical effort may also be a risk factor for stroke. This current analysis could provide an additional argument for or against these relationships. If Authors don’t have a possibilities to perform additional analysis, they should explain the reasons

R2.4. We agree with the reviewer that sedentary lifestyle and physical effort are relevant variables when studying stroke risk factors. However, this part of the analysis tried to reproduce results from a previously published study (Fadel et al. Association Between Reported Long Working Hours and History of Stroke in the CONSTANCES Cohort). Therefore, we included in this analysis only the variables that were used in the aforementioned study. The variables sedentary lifestyle and Borg scales were used in the second analysis which tried to identify risk factors for knee pain.

2.5. In Discussion I found double use of „are more” (stealing data for economic gains are more are more frequent, and health facilities are no longer spared)

R2.5. We have corrected the mistake.

---

## [Editor Report · Decision Letter 2]

20 May 2024

PONE-D-23-27743R2Flexibility of a large blindly synthetized avatar database for occupational research: example from the CONSTANCES cohort for stroke and knee painPLOS ONE

Dear Dr. Fadel,

Thank you for submitting your manuscript to PLOS ONE. After careful consideration, we feel that it has merit but does not fully meet PLOS ONE’s publication criteria as it currently stands. Therefore, we invite you to submit a revised version of the manuscript that addresses the points raised during the review process.

The paper appears to be sound in its content. However, it must be edited to address various grammatical issues. Also, certain sections seemed out of place, and at least one statement was not substantiated with evidence. At this time, I've offered the following recommendations. I would suggest that the authors review the abstract and manuscript and submit a revision. Adjust the references accordingly.

**Abstract:**

Objectives:

Line 1: “Though the rise of big data in the field of occupational health…”

Line 3: ‘…sensitive data from the field of …’

Line 4: ‘…synthesized database developed from the CONSTANCES cohort by comparing associations between three independently selected outcomes, and various exposures.’

Methods:

Line 2: ‘…(Octopize) that is agnostic to the primary or secondary data uses. Three main analyses of interest were chose to compare assocaitions computed in the raw and avatar dataset: risk of stroke (any stroke, and subtypes of stroke), risk of knee pain, and limitations associated with knee pain.’

Results:

Line 2: ‘…more than 77% of the comparisons had….’

**Manuscript:**

Introduction:

Page 3:

Paragraph 1, line 2. Remove the comma “,” after the word “Though”

Paragraph 1, line 6: “sheer number of observations and variables available”

Paragraph 2, line 6: “approaches need a substantial amount of data”

Paragraph 2, line 7: “reproducible” and not “reproductible”

Paragraph 2, Remove the sentence “Likely, data….vaccinations etc. [11, 12]” from paragraph 2 and “One important…sharing.” from paragraph 3.

Paragraph 3:

Begin paragraph 3 as follows: “Issues regarding data security and possible re-identification have grown in the public and governmental agencies. Some studies managed to re-identify databases with great accuracy, which highlights the unsatisfactory reality for data anonymization [13,14]. Data related to the occupational medical files are of particular concern in terms of security and confidentiality as they deal with both health data (medical confidentiality) and data related to the workplace (professional confidentiality). Furthermore, the latest guidelines in France recommend collecting information on sensitive topics like medical fitness for work, occupational exposures, addictions, psychiatric diseases, vaccinations etc. [11,12]. With that in mind, the General Data Protection Regulation (GDPR) imposed harsh conditions for defining anonymous data and proposed several recommendations to evaluate the robustness of an anonymization process.[15]”

Page 4:

Paragraph 1

line 4: “different from the original while retaining its statistical properties”

line 5: “but they often require”

line 6: “and rarely allow”

line 7: “generating synthetic datasets”

line 8: “and has proven to be effective in specific analysis in terms of statistical properties as well as ensuring data privacy”

line 9: “in the context of a large cohort dataset.”

Paragraph 2:

“The aim of this study was to create a large synthetic database and evaluate associations between three selected outcomes and several relevant variables.”

Methods:

Line 3: “adults”

Line 7: “…self-administered follow-up surveys.”

Line 7-9: “…examinations allowed capturing data on demographics, …… pain, and daily life limitations”.

Page 5:

Paragraph 2:

Line 1: The word ‘in’ is used twice. Secondly, say, “both the raw dataset and the avatar dataset”

Line 2: “were computed to evaluate the various associations considered”.

Line 4: What is meant by number of pack years? Do you mean number of packs per year? If so, please state that more clearly.

Line 5: occupational status

Line 6: white-collar jobs and not white jobs. This is mentioned twice in line 6.

Line 14: “The third main analysis assessed the effect of knee pain (as defined in the 2nd main analysis) on several reported limitations: ….”

Pag 6: Line 1: Remove the ellipses (…) after the word “diagnosed”

Table 1 title:

“Distribution of Variables in the Raw and Avatar dataset, Crude Odds Ratios for the Occurrence of Stroke (Stroke Analysis), and Crude Odds Ratios for Reported Limitations, Across Selected Risk Factors”

Table 1: Please state what is meant by Smoking (Pack.Years) more clearly. Does it mean packs per year?

Table 1 footnote: OR for those with moderate knee pain; OR for those with severe knee pain.

Page 8:

Paragraph 1:

The authors state “Though most of the number in each variable were statistically different….”, there is not statistical analysis reported to justify this statement. The only statement that can be made is that “Differences between the Avatar and Raw data sets in terms of the percentage distribution of responses for the various risk factors were mostly small (≤2%), with only one category…..10%).

Discussion:

Line 2 - 5: “The aim of this study was…..with minimum assumptions about the proposed analysis.”

Line 6: “…for economic gain has become more frequent, and health facilities are not immune to this threat.”

Page 9:

Line 2: “…exploratory analysis.”

Line 3: researchers and not “searchers”.

Line 5: “…relevant in the case of data pertaining to work, …”

Line 6: “black-box machine learning…”

Line 10: I don’t understand what is meant by “restitution of statistical analysis to employers and workers”

Line 11: “…used for analysis with employers…”

Line 12: “...multicentric analysis (e.g. across occupational health…)”

Last paragraph: “The main limitation of this study…”

Last line: “…which is not available in open access.”

Page 10:

Line 1: “…was made since the aim of this study was to…”

Line 2-3: “Most studies compare the overall of confidence intervals,[24] though this method has limitations in the case of large datasets with small confidence intervals, as was the case our analysis of knee pain, which had close OR with non-overlapping confidence intervals.”

Line 9: “…from rare events like stroke, to common…”

We look forward to receiving your revised manuscript.

Kind regards,

Mir Ali, Ph.D

Guest Editor

PLOS ONE

Journal Requirements:

Additional Editor Comments:

The paper appears to be sound in its content. However, it has to be edited to address various grammatical issues. Also, certain sections seemed out of place, and at least one statement was not substantiated with evidence. At this time, I've offered the following recommendations. I would suggest that the authors review the abstract and manuscript and submit a revision. Adjust the references accordingly.

Abstract:

Objectives:

Line 1: “Though the rise of big data in the field of occupational health…”

Line 3: ‘…sensitive data from the field of …’

Line 4: ‘…synthesized database developed from the CONSTANCES cohort by comparing associations between three independently selected outcomes, and various exposures.’

Methods:

Line 2: ‘…(Octopize) that is agnostic to the primary or secondary data uses. Three main analyses of interest were chose to compare assocaitions computed in the raw and avatar dataset: risk of stroke (any stroke, and subtypes of stroke), risk of knee pain, and limitations associated with knee pain.’

Results:

Line 2: ‘…more than 77% of the comparisons had….’

Manuscript:

Introduction

Page 3:

Paragraph 1, line 2. Remove the comma “,” after the word “Though”

Paragraph 1, line 6: “sheer number of observations and variables available”

Paragraph 2, line 6: “approaches need a substantial amount of data”

Paragraph 2, line 7: “reproducible” and not “reproductible”

Paragraph 2, Remove the sentence “Likely, data….vaccinations etc. [11, 12]” from paragraph 2 and “One important…sharing.” from paragraph 3.

Paragraph 3:

Begin paragraph 3 as follows: “Issues regarding data security and possible re-identification have grown in the public and governmental agencies. Some studies managed to re-identify databases with great accuracy, which highlights the unsatisfactory reality for data anonymization [13,14]. Data related to the occupational medical files are of particular concern in terms of security and confidentiality as they deal with both health data (medical confidentiality) and data related to the workplace (professional confidentiality). Furthermore, the latest guidelines in France recommend collecting information on sensitive topics like medical fitness for work, occupational exposures, addictions, psychiatric diseases, vaccinations etc. [11,12]. With that in mind, the General Data Protection Regulation (GDPR) imposed harsh conditions for defining anonymous data and proposed several recommendations to evaluate the robustness of an anonymization process.[15]”

Page 4:

Paragraph 1

line 4: “different from the original while retaining its statistical properties”

line 5: “but they often require”

line 6: “and rarely allow”

line 7: “generating synthetic datasets”

line 8: “and has proven to be effective in specific analysis in terms of statistical properties as well as ensuring data privacy”

line 9: “in the context of a large cohort dataset.”

Paragraph 2:

“The aim of this study was to create a large synthetic database and evaluate associations between three selected outcomes and several relevant variables.”

Methods:

Line 3: “adults”

Line 7: “…self-administered follow-up surveys.”

Line 7-9: “…examinations allowed capturing data on demographics, …… pain, and daily life limitations”.

Page 5:

Paragraph 2:

Line 1: The word ‘in’ is used twice. Secondly, say, “both the raw dataset and the avatar dataset”

Line 2: “were computed to evaluate the various associations considered”.

Line 4: What is meant by number of pack years? Do you mean number of packs per year? If so, please state that more clearly.

Line 5: occupational status

Line 6: white-collar jobs and not white jobs. This is mentioned twice in line 6.

Line 14: “The third main analysis assessed the effect of knee pain (as defined in the 2nd main analysis) on several reported limitations: ….”

Pag 6: Line 1: Remove the ellipses (…) after the word “diagnosed”

Table 1 title:

“Distribution of Variables in the Raw and Avatar dataset, Crude Odds Ratios for the Occurrence of Stroke (Stroke Analysis), and Crude Odds Ratios for Reported Limitations, Across Selected Risk Factors”

Table 1: Please state what is meant by Smoking (Pack.Years) more clearly. Does it mean packs per year?

Table 1 footnote: OR for those with moderate knee pain; OR for those with severe knee pain.

Page 8:

Paragraph 1:

The authors state “Though most of the number in each variable were statistically different….”, there is not statistical analysis reported to justify this statement. The only statement that can be made is that “Differences between the Avatar and Raw data sets in terms of the percentage distribution of responses for the various risk factors were mostly small (≤2%), with only one category…..10%).

Discussion:

Line 2 - 5: “The aim of this study was…..with minimum assumptions about the proposed analysis.”

Line 6: “…for economic gain has become more frequent, and health facilities are not immune to this threat.”

Page 9:

Line 2: “…exploratory analysis.”

Line 3: researchers and not “searchers”.

Line 5: “…relevant in the case of data pertaining to work, …”

Line 6: “black-box machine learning…”

Line 10: I don’t understand what is meant by “restitution of statistical analysis to employers and workers”

Line 11: “…used for analysis with employers…”

Line 12: “...multicentric analysis (e.g. across occupational health…)”

Last paragraph: “The main limitation of this study…”

Last line: “…which is not available in open access.”

Page 10:

Line 1: “…was made since the aim of this study was to…”

Line 2-3: “Most studies compare the overall of confidence intervals,[24] though this method has limitations in the case of large datasets with small confidence intervals, as was the case our analysis of knee pain, which had close OR with non-overlapping confidence intervals.”

Line 9: “…from rare events like stroke, to common…”

---

## [Author Response · Author response to Decision Letter 2]

1 Jul 2024

Abstract:

Objectives:

Line 1: “Though the rise of big data in the field of occupational health…”

R0.1. Corrected.

Line 3: ‘…sensitive data from the field of …’

R0.2. Corrected.

Line 4: ‘…synthesized database developed from the CONSTANCES cohort by comparing associations between three independently selected outcomes, and various exposures.’

R0.3. Corrected.

Methods:

Line 2: ‘…(Octopize) that is agnostic to the primary or secondary data uses. Three main analyses of interest were chose to compare assocaitions computed in the raw and avatar dataset: risk of stroke (any stroke, and subtypes of stroke), risk of knee pain, and limitations associated with knee pain.’

R0.4. Corrected.

Results:

Line 2: ‘…more than 77% of the comparisons had….’

R0.5. Corrected.

Manuscript:

Introduction:

Page 3:

Paragraph 1, line 2. Remove the comma “,” after the word “Though”

R0.6. Corrected.

Paragraph 1, line 6: “sheer number of observations and variables available”

R0.7. Corrected.

Paragraph 2, line 6: “approaches need a substantial amount of data”

R0.8. Corrected.

Paragraph 2, line 7: “reproducible” and not “reproductible”

R0.9. Corrected.

Paragraph 2, Remove the sentence “Likely, data….vaccinations etc. [11, 12]” from paragraph 2 and “One important…sharing.” from paragraph 3.

R0.10. Corrected.

Paragraph 3:

Begin paragraph 3 as follows: “Issues regarding data security and possible re-identification have grown in the public and governmental agencies. Some studies managed to re-identify databases with great accuracy, which highlights the unsatisfactory reality for data anonymization [13,14]. Data related to the occupational medical files are of particular concern in terms of security and confidentiality as they deal with both health data (medical confidentiality) and data related to the workplace (professional confidentiality). Furthermore, the latest guidelines in France recommend collecting information on sensitive topics like medical fitness for work, occupational exposures, addictions, psychiatric diseases, vaccinations etc. [11,12]. With that in mind, the General Data Protection Regulation (GDPR) imposed harsh conditions for defining anonymous data and proposed several recommendations to evaluate the robustness of an anonymization process.[15]”

R0.11. Corrected.

Page 4:

Paragraph 1

line 4: “different from the original while retaining its statistical properties”

R0.12. Corrected.

line 5: “but they often require”

R0.13. Corrected.

line 6: “and rarely allow”

R0.14. Corrected.

line 7: “generating synthetic datasets”

R0.15. Corrected.

line 8: “and has proven to be effective in specific analysis in terms of statistical properties as well as ensuring data privacy”

R0.16. Corrected.

line 9: “in the context of a large cohort dataset.”

R0.17. Corrected.

Paragraph 2:

“The aim of this study was to create a large synthetic database and evaluate associations between three selected outcomes and several relevant variables.”

R0.18. Corrected.

Methods:

Line 3: “adults”

R0.19. Corrected.

Line 7: “…self-administered follow-up surveys.”

R0.20. Actually, the questionnaire were all retrieved at the baseline inclusion.

Line 7-9: “…examinations allowed capturing data on demographics, …… pain, and daily life limitations”.

R0.21. Corrected.

Page 5:

Paragraph 2:

Line 1: The word ‘in’ is used twice. Secondly, say, “both the raw dataset and the avatar dataset”

R0.22. Corrected.

Line 2: “were computed to evaluate the various associations considered”.

R0.23. Corrected.

Line 4: What is meant by number of pack years? Do you mean number of packs per year? If so, please state that more clearly.

R0.24. Pack-years refer to the number of packs of cigarette a person has smoked per day per year. It is obtained by multiplying the mean number of packs smoked per day by the number of years the person has smoked (for example 1 pack/day smoked for 10 years corresponds to 10 pack-years).

This point was clarified in the methods:

P5. “(number of pack years, i.e. the number of packs smoked per day multiplied by the number of years the person has smoked)”

Line 5: occupational status

R0.25. Corrected.

Line 6: white-collar jobs and not white jobs. This is mentioned twice in line 6.

R0.26. Corrected.

Line 14: “The third main analysis assessed the effect of knee pain (as defined in the 2nd main analysis) on several reported limitations: ….”

R0.27. Corrected.

Pag 6: Line 1: Remove the ellipses (…) after the word “diagnosed”

R0.28. Corrected.

Table 1 title:

“Distribution of Variables in the Raw and Avatar dataset, Crude Odds Ratios for the Occurrence of Stroke (Stroke Analysis), and Crude Odds Ratios for Reported Limitations, Across Selected Risk Factors”

R0.29. Corrected.

Table 1: Please state what is meant by Smoking (Pack.Years) more clearly. Does it mean packs per year?

R0.30. A definition of pack-years was added in the method section.

Table 1 footnote: OR for those with moderate knee pain; OR for those with severe knee pain.

R0.31. Corrected.

Page 8:

Paragraph 1:

The authors state “Though most of the number in each variable were statistically different….”, there is not statistical analysis reported to justify this statement. The only statement that can be made is that “Differences between the Avatar and Raw data sets in terms of the percentage distribution of responses for the various risk factors were mostly small (≤2%), with only one category…..10%).

R0.32. As we do not show statistical tests, we have amended this part as recommended.

Discussion:

Line 2 - 5: “The aim of this study was…..with minimum assumptions about the proposed analysis.”

R0.33. Corrected.

Line 6: “…for economic gain has become more frequent, and health facilities are not immune to this threat.”

R0.34. Corrected.

Page 9:

Line 2: “…exploratory analysis.”

R0.35. Corrected.

Line 3: researchers and not “searchers”.

R0.36. Corrected.

Line 5: “…relevant in the case of data pertaining to work, …”

R0.37. Corrected.

Line 6: “black-box machine learning…”

R0.38. Corrected.

Line 10: I don’t understand what is meant by “restitution of statistical analysis to employers and workers”

R0.39. What we meant to say is that synthetize data can be presented to employers and workers without risking privacy breaches. This point was clarified in the discussion:

P9. “Possible application examples could include presentation of data to employers and workers on sensitive issues (like addictions), sharing databases used for analyses (transparency), sharing databases for multicentric analyses (e.g. across occupational health units) with possibilities of international projects, or sharing databases that were used to train machine learning models”

Line 11: “…used for analysis with employers…”

R0.40. Corrected (see R0.39).

Line 12: “...multicentric analysis (e.g. across occupational health…)”

R0.41. Corrected (see R0.39).

Last paragraph: “The main limitation of this study…”

R0.42. Corrected.

Last line: “…which is not available in open access.”

R0.43. Corrected.

Page 10:

Line 1: “…was made since the aim of this study was to…”

R0.44. Corrected.

Line 2-3: “Most studies compare the overall of confidence intervals,[24] though this method has limitations in the case of large datasets with small confidence intervals, as was the case our analysis of knee pain, which had close OR with non-overlapping confidence intervals.”

R0.45. Corrected.

Line 9: “…from rare events like stroke, to common…”

R0.46. Corrected.

---

## [Editor Report · Decision Letter 3]

17 Jul 2024

Flexibility of a large blindly synthetized avatar database for occupational research: example from the CONSTANCES cohort for stroke and knee pain

PONE-D-23-27743R3

Dear Dr. Fadel,

We’re pleased to inform you that your manuscript has been judged scientifically suitable for publication and will be formally accepted for publication once it meets all outstanding technical requirements.

Kind regards,

Mir Ali, Ph.D

Guest Editor

PLOS ONE
---

## [Editor Report · Acceptance letter]

22 Jul 2024

PONE-D-23-27743R3 

PLOS ONE

Dear Dr. Fadel, 

I'm pleased to inform you that your manuscript has been deemed suitable for publication in PLOS ONE. Congratulations! Your manuscript is now being handed over to our production team.

Kind regards, 

on behalf of

Dr. Mir Ali 

Guest Editor

PLOS ONE